

# Total ozone variability and trends over the South Pole during the wintertime

Vitali Fioletov[1], Xiaoyi Zhao[1], Ihab Abboud[1], Michael Brohart[1], Akira Ogyu[1], Reno Sit[1], Sum Chi Lee[1], Irina Petropavlovskikh[2,3], Koji Miyagawa[2], Bryan J. Johnson[2], Patrick Cullis[2], John Booth[2,*], Glen McConville[2,3], and C. Thomas McElroy[4]

[1]Air Quality Research Division, Environment and Climate Change Canada, Toronto, Ontario, M3H 5T4, Canada.
[2]Global Monitoring Laboratory Earth System Research Laboratory, NOAA, Boulder, CO, 80305 USA
[3]Cooperative Institute for Research in Environmental Sciences, University of Colorado, Boulder, CO, USA
[4]York University, Toronto, Ontario, M3J 1P3, Canada.
[*]Deceased

*Correspondence to*: Vitali Fioletov (vitali.fioletov@outlook.com or vitali.fioletov@ec.gc.ca)

**Abstract.** The Antarctic polar vortex creates unique chemical and dynamical conditions when the stratospheric air over Antarctica is isolated from the rest of the stratosphere. As a result, stratospheric ozone within the vortex remains largely unchanged for a five-month period from April until late August when the sunrise and extremely cold temperatures create favorable conditions for rapid ozone loss. Such prolonged stable conditions within the vortex make it possible to estimate the total ozone levels there from sparse wintertime ozone observations at the South Pole. The available records of focused Moon (FM) observations by Dobson and Brewer spectrophotometers at the Amundsen-Scott South Pole Station (for the periods 1964–2022 and 2008–2022, respectively) as well as integrated ozonesonde profiles (1986–2022) and MERRA-2 reanalysis data (1980–2022) were used to estimate the total ozone variability and long-term changes over the South Pole. Comparisons with MERRA-2 reanalysis data for the period 1980–2022 demonstrated that the uncertainties of Dobson and Brewer daily mean FM values are about 2.5%–4%. Wintertime (April-August) MERRA-2 data have a bias with Dobson data of -8.5% in 1980–2004 and 1.5% in 2005–2022. The mean difference between wintertime Dobson and Brewer data in 2008-2022 was about 1.6%; however, this difference can be largely explained by various systematic errors in Brewer data. The wintertime ozone values over the South Pole during the last 20 years were about 12% below the pre-1980s level, i.e., the decline there was nearly twice larger than that over southern midlatitudes. It is probably the largest long-term ozone decline aside from the springtime Antarctic ozone depletion. While wintertime ozone decline over the pole has hardly any impact on the environment, it can be used as an indicator to diagnose the state of the ozone layer, particularly because it requires data from only one station. Dobson and ozonesonde data after 2001 show a small positive, but not statistically significant, trend in ozone values of about 1.5% per decade that is in line with the trend expected from the concentration of the ozone depleting substances in the stratosphere.



# 1 Introduction

The wintertime stratosphere circulation is dominated by a large cyclonic vortex centered near the pole. A very strong polar vortex in the Antarctic stratosphere creates unique chemical and dynamic conditions and isolates stratospheric air over Antarctica from the rest of the stratosphere (Nash et al., 1996). It forms in austral autumn, reaches maximum strength in midwinter, and stays up to December. The variability of the Antarctic vortex is small (Waugh and Randel, 1999) except during the spring vortex breakdown in late spring/summer, although there are some rare exceptions of earlier vortex disruptions. For example, the vortex broke up in September 2002 (Allen et al., 2003; Hoppel et al., 2003; Ricaud et al., 2005) and it demonstrated large disturbance as early as late August in 1988 (Johnson et al., 2023) and 2019 (Wargan et al., 2020; Safieddine et al., 2020; Milinevsky et al., 2020). Nevertheless, for a period of five months, from April to late August, there is a strong undisturbed vortex over the South Pole where ozone is isolated from the rest of the stratosphere and therefore remained relatively unchanged during the wintertime.

The southern hemisphere springtime polar vortex has been a subject of intense research since the discovery and subsequent studies of the Antarctic ozone hole (Farman et al., 1985; Solomon et al., 1986; Stolarski et al., 1986). In austral spring, a unique combination of cold temperature, sunlight, polar stratospheric clouds, and a substantial concentration of ozone-depleting substances over Antarctica led to a very rapid destruction of ozone in the lower stratosphere (e.g., Solomon et al., 1986; WMO, 2018; 2022, and references therein). The wintertime ozone in the vortex is less studied. First, because no photochemical ozone destruction processes occur during the polar night, and therefore, no rapid changes in ozone are expected. Second, because the most reliable satellite ozone instruments, for example, the Solar Backscatter Ultraviolet Radiometer (e.g., McPeters et al., 2013), derive column ozone from backscattered solar radiation and therefore cannot measure wintertime column ozone during the polar night.

Long-term ozone trends over the South Pole in the wintertime should be a good indication of the ozone layer state. There is a strong latitudinal gradient toward the pole in total ozone declining trends in the southern hemisphere from a near-zero trends over the equator to a total decline of about 8% at 60°S between 1979 and 1996 (e.g., Vyushin et al., 2007; Weber et al., 2018) with little dependence of the season except for the springtime ozone hole (e.g., Fioletov et al., 2002). Similarly, the ozone recovery trends after 1996 are also increasing from the equator toward high latitudes (Weber et al., 2022). Therefore, from such latitudinal gradient in the trends, it can be expected that the wintertime ozone changes are the largest over the South Pole.

Unlike the North Pole, the South Pole is practically always under the Antarctic polar vortex during winter (Karpetchko et al., 2005; Waugh and Polvani, 2010). Therefore, measurements from only one location, the Amundsen-Scott South Pole Station, can provide information on the state of the wintertime ozone layer within the polar vortex. Wintertime total ozone measurements at the South Pole Station are available from Dobson and Brewer spectrophotometers that use the Moon as the light source (Komhyr et al., 1988; McElroy et al., 2010) as well as from ozonesondes, where total ozone can be obtained by integrating the ozone profile (e.g., Johnson et al., 2023).



Information about wintertime ozone over the South Pole Station is also available from the recent Modern-Era Retrospective Analysis for Research and Applications, Version 2 (MERRA-2) reanalysis (Gelaro et al., 2017). The advantage of MERRA-2 is that it provides a continuous record for the period 1980–2022 with an hourly temporal resolution. Every Dobson and Brewer measurement and ozonesonde flight can be matched with a nearly coincident MERRA-2 value to identify potential problems with data and study sampling effects.

The paper is organized as follows: Section 2 describes the datasets used: ground-based ozone measurements by the Dobson and Brewer instruments, total ozone from integrated ozonesondes, and the MERRA-2 reanalysis total ozone data. Section 3 discusses available datasets, differences between them and possible corrections for the data. Wintertime total ozone variability and long-term changes are discussed in Section 4. Discussion and conclusions are given in Section 5.

## 2 Instruments and datasets

### 2.1 Dobson spectrophotometer, 1964-2022

The Dobson spectrophotometer was developed in the 1920s and continuous regular measurements started, first in Europe in the 1920s (Dobson and Harrison, 1926, Dobson, 1968), and later in other parts of the globe (Brönnimann et al., 2003). In Antarctica, Dobson measurements started during the International Geophysical year in 1957–1958. About 130 instruments were produced and about 50 Dobson stations remain operational today (Fioletov et al., 2008) including 6 in Antarctica. Antarctic Dobson measurements led to a discovery of large springtime ozone depletion over Antarctica, or the Ozone Hole (Farman et al., 1985; Bhartia and McPeters, 2018): anomalous total ozone behaviour was uncovered from Dobson measurements at the Japanese station Syowa (Chubachi, 1984) and British station Hally Bay (Farman et al., 1985). Dobson measurements at the South Pole Station started in late 1963. It turns out that some of the early Dobson data recorded there were incorrect, presumably caused by operator error, and were later retracted and reprocessed (Komhyr et al., 1986; Bhartia and McPeters, 2018). Large springtime Antarctic ozone depletion was then further confirmed (Komhyr et al, 1988; 1989b).

The Dobson instrument description, information on its uncertainties, calibration and operation procedures can be found in several publications (e.g., Komhyr, 1980a; Basher, 1982; Komhyr et al., 1989a; Komhyr and Evans, 2008). The instrument uses three wavelength pairs designated as – A (A1:305.5/A2:325.0 nm), – C (C1:311.5/C2:332.4 nm), – D (D1:317.5/D2:339.9 nm) and double-pair combinations (typically, AD and CD) are used to retrieve total ozone to minimize the optical effect of atmospheric aerosols. The resultant ozone from the AD and CD combinations do not always agree and instructions to account for this difference are described in the standard operating procedures (Komhyr and Evans, 2008). A well-known source of potential biases in Dobson measurements is related to the effects of temperature and vertical ozone profile on the derived Dobson total ozone. It is because the standard Dobson retrievals are based on the assumption of a standard stratospheric temperature of -46.3°C and a standard ozone profile (Komhyr et al., 1993). In the case of the South Pole Station, such assumptions could lead to up to 4% systematic errors (Bernhard et al., 2005). Redondas et al., 2014 suggested





a correction for Dobson systematic errors that was applied by Evans et al., 2017 and such a corrected version of South Pole data was used in this study (see also https://gml.noaa.gov/aftp/data/ozwv/Dobson/Publications/, last access: 10 April 2023, for details). The correction is based on the effective temperature climatology calculated from the ozone and temperature climatological profiles by McPeters and Labow, 2012. Note that this version is available from NOAA and is different from the version, available from the World Ozone and Ultraviolet Radiation Data Centre (WOUDC, http://woudc.org, last access: 10 February 2023).

Dobson measurements at the South Pole Station are taken using the direct irradiance from the Sun (DS), the Moon (FM), or scattered light from the Zenith Sky (ZS). Only DS and FM measurements are used in this study. There is typically 10-30 measurement per day and one of them is reported as a representative daily value. Most of the South Pole measurements were taken by Dobson instruments 80 and 82 with a short period of measurements by Dobson 42 as shown in Figure 1. All these Dobsons are calibrated against the world primary Dobson standard instrument 83 (Komhyr et al., 1989a).

## 2.2 Brewer spectrophotometer, 2008-2022

The Brewer instrument was proposed by Alan Brewer (Brewer, 1973) and developed in the early 1980s at Environment and Climate Change Canada (ECCC) (Kerr, 2010; Kerr et al., 1981). Unlike the original Dobson instrument, it is a fully automated instrument that can take FM ozone measurements approximately every 15 minutes. More than 230 Brewer instruments have been manufactured and 88 Brewer stations have reported their data to the WOUDC (Zhao et al., 2021) including 9 in Antarctica. ECCC Brewer no. 085 was installed, under an agreement between ECCC and the US National Oceanic and Atmospheric Administration (NOAA), in February of 2008 (McElroy et al., 2012). It was replaced by ECCC Brewer no. 021 in December 2014 as shown in Figure 1.

There are two main types of Brewer instruments. The originally developed instruments were single spectrophotometers (types Mark II and IV). The double monochromator (Mark III) was introduced in the early 1990s to reduce stray light and to enable accurate ozone measurements under the low sun conditions (Wardle et al., 1996) which is particularly important for high-latitude sites. The Mark III uses the same concept as the Mark II model but has a second spectrometer. Both Brewer instruments operated at the South Pole Station were Mark III type. All ECCC Brewers are calibrated against the Brewer world primary standard (the Brewer triad) at Toronto (Fioletov et al., 2005; Zhao et al., 2021).

The Brewer spectrophotometer is a modified Ebert grating spectrometer that measures the intensity of radiation at six selected channels in UV (303.2, 306.3, 310.1, 313.5, 316.8, and 320.1 nm), where the four longer wavelengths are used to retrieve total column ozone. Similar to the Dobson instrument, the Brewer can perform ozone measurements in the DS, ZS, and FM modes. Details about the Brewer instrument, retrieval algorithm, instrument operation and calibration can be found in a recent overview by Kerr, (2010). Similar to the Dobson, the Brewer retrieval algorithm is using Bass and Paur ozone absorption cross-sections interpolated to a standard stratospheric temperature of -45°C (Bassand Paur, 1985). Brewer ozone retrievals are much less affected by stratospheric temperatures than Dobsons (Kerr, 2002; Redondas et al., 2014; Gröbner,



2021). We estimated that the errors in retrieved ozone, introduced by the stratospheric temperatures over the South pole, are under 0.4%.

ECCC Brewers at the South Pole Station typically perform 3-4 FM measurement per hour during the polar night and 5-6 DS and ZS measurements per hour during the polar day. FM data were screened out if the standard error of individual measurements exceeded 12 Dobson Units (DU), if the lunar disc illumination was less than 50%, or if the lunar zenith angle exceeds 76°. Brewer measurements at the South Pole Station are available from the WOUDC.

## 2.3 Ozonesondes, 1986-2022

Regular balloon-borne ozonesondes providing high-resolution vertical profiles of ozone and temperature at the South Pole Station started in 1986 (Hofmann et al., 1997; 2009; Solomon et al., 2005). There is typically one flight per week, although the frequency is often higher (2–3 flights per week) during the ozone hole period. The electrochemical concentration cell (ECC) ozonesondes were used for the entire period and their design (Komhyr, 1967) has remained relatively unchanged throughout the whole record. During the cold months (from April to mid-October), polyethylene film balloons were used assuring burst altitudes of about 30 km. Standard rubber balloons were used for other months. The entire South Pole ozonesonde record has been harmonised by Sterling et al. (2018). An overall review of the record and estimates of ozone variability and trends for the ozone hole period are available from a recent paper by Johnson et al., (2023).

To obtain total ozone from an integrated ozonesonde profile, it is necessary to make assumptions about the ozone profile above the balloon burst height. There are two approaches developed for extrapolation of the ozone profile: (1) assuming a constant mixing ratio (CMR)of ozone above the balloon burst pressure (~ 20 to 7 hPa) to zero pressure or (2) reconstructing the missing part of the profile using the satellite Solar Backscatter Ultraviolet Radiometer (SBUV) climatology (McPeters et al., 1997). Total ozone calculated by both methods are available from the data sets developed by NOAA. Following Wargan et al., (2017) and Johnson et al., (2023), the dataset used in this study is the one where the interpolation is done by the first method. As noted by Johnson et al., (2023), "the CMR extrapolation is more suitable over South Pole during the polar night and low sun angle months when satellite and ground-based optical measurements are limited". We have found that the difference in estimated total ozone long-term variation between the values estimated using the two methods are rather minor. Johnson et al., (2023) also reported that after the homogenization, there is a constant 2 ± 3% offset: ozonesonde total ozone is slightly higher than the DS Dobson observations.

## 2.4 MERRA-2 reanalysis data, 1980-2022

The second Modern-Era Retrospective analysis for Research and Applications (MERRA-2) is an atmospheric reanalysis from NASA's Global Modeling and Assimilation Office (Gelaro et al., 2017). MERRA-2 assimilates partial column ozone retrievals from the NOAA SBUV/2 series (nos. 11, 14, 16, 17, 18, 19) from 1980 to 2004. From October 2004, MERRA-2 assimilate ozone profiles the Microwave Limb Sounder (MLS) and total column data from and the Ozone Monitoring Instrument (OMI)



(Wargan et al., 2017). Both OMI and MLS instruments are onboard of the Earth Observing System Aura satellite that was launched in 2004. While SBUV and OMI instruments measure ozone using solar light backscattered by the atmosphere, MLS observes thermal microwave emission from Earth's limb and its stratospheric ozone mixing ratio data are available during the polar night up to 82° of latitude (Wargan et al., 2017). The MERRA-2 ozone record was divided into two periods, herein referred to as the SBUV period and the Aura period. MERRA-2 total ozone is a continuous record from 1980 to 2022 with 1-hour resolution. MERRA-2 ozone data has been found to have good quality when compared with satellite and ground-based observations (e.g., Rienecker et al., 2011; Wargan et al., 2017; Zhao et al., 2017, 2019, 2021). For exemple, in Zhao et al. (2021), the bias between Brewer world reference instruments and MERRA-2 is from -0.27% to 1.05% (hourly data; 1999–2019 period), with a monthly difference's standard deviation less than 1.2%.

Wargan et al., (2017) evaluated the MERRA-2 ozone fields using ground-based data including total ozone data from integrated ozonesonde profiles over the South Pole. They found that during both the SBUV and Aura periods, MERRA-2 is lower than the ozonesondes by 3%, and the standard deviation of difference between MERRA-2 and ozonesondes is 12% in the SBUV period and only 5% in the Aura period. Note that these numbers represent the estimates for the entire year, although Wargan et al., (2017) noted the existence of some systematic seasonal biases.

## 3. Comparison of the data records

The time series of total ozone from the four data sources for the entire period of observations are shown in Figure 1. The ozone hole formation is clearly visible on the plot, but it also shows that wintertime FM measurements by Dobsons and Brewers were taken almost every year and a large number of such measurements are available. Note that while the solar zenith angles vary in the same range every year, the span of the lunar zenith angles is different from year to year (Figure 1e) with the minimum value varying from about 62° and to about 72°. There are typically 5-6 periods during winter, when the Moon is nearly full and the lunar zenith angles are the smallest (see Appendix A for details). As the range of lunar zenith angles slowly varies from year to year, artificial long-term changes in total ozone could be introduced if an instrument has a lunar zenith angle-dependent error.

To illustrate short-term ozone fluctuations and the measurement availability, Figure 2 (top) shows total ozone values from the four data sources for two weeks in 2016 along with the lunar zenith angles and lunar disk illumination plots. While DS measurements at the South Pole Station are available almost every day in summer, the number of days with good-quality FM measurements per month is only 4–7 (27 days of good FM nights per winter on average). In the example shown in Figure 2 (top), there are two 4-day periods with continuous Brewer FM measurements when the Moon was nearly full, while Dobson data are available once a day, and ozonesonde data are available once a week. Although Brewer measurements show high scattering, both Brewer and MERRA-2 data demonstrate similar ozone fluctuations and the correlation coefficient between them is about 0.8. MERRA-2 data captured the rapid ozone changes on June 20–22 very well, although the peak on July 16-17, which is seen in both Brewer and Dobson data, does not appear to the same extent in MERRA-2. Brewer measurements



on July 18-20 show some diurnal variations that are not seen in MERRA-2. This could be related to some horizontal inhomogeneity of the ozone distribution over the pole that led to variations in measured ozone due to changing lunar azimuth angle. As Figure 2 (middle and bottom) shows, the Dobson and Brewer measurements on the plot are taken during the optimal periods when the Moon was full, and the zenith angles were near the minimum.

MERRA-2 data can be matched with every Dobson, Brewer, and integrated total ozone measurement since 1980 and then differences with MERRA-2 can be used to analyse potential biases among the four data sets. Figure 3 (left column) shows the differences with MERRA-2 as a function of year and month of the year for the polar night (April–August) and polar day (October-February). March and September were excluded from the analysis since the number of Dobson and Brewer measurements during the sunrise and sunset months is very limited. It is expected that MERRA-2 total ozone would have

different characteristics during the SBUV period (1980–2004) and the Aura period (2005–2022) and Figure 3 shows the comparison results for these two periods separately.

        There are several discrepancies between the four analyzed total ozone data sets during the wintertime. Some adjustments were applied to remove these discrepancies as described below. Dobson total ozone data were thoroughly analyzed in the past (e.g., Bernhard et al., 2005) and no adjustments were applied. However, 3 unrealistic FM ozone values below 200

DU in 2013, that were very different from the Brewer, ozonesonde, and MERRA-2 values, were deleted. As mentioned in Section 2.3, there is a 2% bias (ozonesonde values are higher) between Dobson DS measurement and integrated ozonesonde total ozone (Johnson et al., 2023). The bias slightly depends on the analysed time interval and season. We used a 2% value for the bias and all integrated total ozone values from ozonesonde flights were reduced by that amount.

        MERRA-2 data during the SBUV period do not have any ozone measurements in the polar night area that can be used

for data assimilation. Not surprisingly, there are a noticeable difference between the SBUV and Aura periods in wintertime MERRA-2 total ozone over the South Pole. There are two approaches to estimate that difference. First, Dobson data can be used as "true" values and the difference in ozone between the SBUV and Aura periods can be calculated from comparison with Dobson. Second, the switch from SBUV to Aura occurred in late 2004, i.e., near the maximum of stratospheric chlorine loading that occurred over Antarctica in 2000-2001 (Newman et al., 2007). Therefore, it can be expected that the ozone levels

in the wintertime polar vortex remain approximately the same during a few years before and after the switch in 2004. Both these approaches give approximately the same differences, and based on these estimates, all April-August MERRA-2 data for 1980-2004 were increased by 8.5% and data for 2005-2022 were decreased by 1.7%. This correction removed a jump in the MERRA-2 record in 2004 and the 10-year averages prior to and after the switch in 2004 became nearly identical. The mean April-August values for 1995–2004 and 2005–2014 were equal to 245 DU from both Dobson and adjusted ozonesonde data.

The same values were 225 and 250 DU for original MERRA-2 corresponding to 245 and 246 DU for corrected MERRA-2. There was also a 3% difference between MERRA-2 and Dobson data in October-February 1980–2004. MERRA-2 data were adjusted to remove that bias for some of the plots. The mean difference between Dobson and MERRA-2 was less than 1% during October-February 2005–2022 and no adjustment was applied to these data.



For Brewer DS measurements, there is a good overall agreement with Dobson data with the mean difference of 0.9% and standard deviation of the difference for daily values of 4%. There was also good agreement between Brewers 085 and 021 DS data during a two-month period (December 2014-January 2015) where both instruments were at the South Pole Station: the mean difference was 0.4% and the standard deviation of the difference was 0.5%. For Brewer FM measurements, uncertainties and systematic errors are larger than for DS data. The mean Brewer – Dobson difference in April-August is 1.6% and the standard deviation for daily values is 7.3%. Thus, on average, independent Brewer FM measurements without any adjustments report total ozone values that are similar to those from the Dobson (within 1.6%). There are, however, several sources of systematic errors in Brewer FM data that led to drifts in Brewer FM ozone values.

A detailed analysis of systematic errors of Brewers nos. 021 and 085 FM measurements is given in the Appendix B. Both Brewers tend to overestimate ozone when lunar direct irradiance at 320 nm is low and when the lunar zenith angle is high. The overestimation is as large as 10%–15% when the ozone slant path is large (greater than 1000 DU). The latter leads to an artificial long-term change in wintertime total ozone due to long-term changes in the lunar zenith angle (Figure 1). Corrections for these two factors were introduced assuming that the data at the higher lunar irradiance and the lowest lunar zenith angles are accurate as discussed in Appendix B.

Figure 3 shows differences of Dobson, Brewer, and ozonesonde total ozone with MERRA-2 for the adjusted data (the right panel) for the same two periods and two seasons for the original data (left panel). The bias in MERRA-2 in April–August 1980–2004 is largely removed and ozonesonde data no longer show a difference with respect to the Dobson data anymore. From now on, the adjusted data are used in this study, unless it is specifically stated otherwise. Note that these corrected Brewer data were also used in Figure 2.

This study is focused on the period from April to August because the vortex is stable during that time and ozone is relatively unchanged, so its characteristics could be estimated from a limited number of measurements. This is further illustrated by Figure 4 where the total ozone annual cycle for three periods is shown. The long-term monthly means in each of these five months are nearly identical. Only the August values were slightly lower in recent years because ozone depletion in the polar vortex starts in late August (e.g., Hassler et al., 2011). For this reason, data for August 20–31 were excluded from the analysis below. Figure 4 shows monthly mean values calculated from all available data for that month. Some differences between individual data sets in September-December are caused by the sampling bias. The number of Dobson and Brewer measurements in March and September is very limited and the measurements are not available in the second half of September. Ozonesonde flights were more frequent when the ozone hole was over the South Pole. There is no sampling bias in April-August, although MERRA-2 data were available every day, while ozonesonde are flown 4–5 times a month and Dobson and Brewer measurements were taken 5–7 days per month.

It is challenging to make Dobson and Brewer measurements during the polar night and such measurements are subject to considerable uncertainties. Nevertheless, the correlation coefficients in April-August between MERRA-2 daily means and Dobson, Brewer, and ozonesonde total ozone daily values were all about 0.8 during the Aura period (2005-2022). The





correlation coefficient between Brewer and Dobson measurements is slightly lower, about 0.7, suggesting that there is some noise in these measurements. For the MERRA-2 SBUV period (1980-2004), the correlation with Dobson and ozonesonde data was much lower, only about 0.4. The correlation coefficient between ozonesonde and Dobson values for 1986-2004 was 0.55 compared to 0.83 for 2005-2022 suggesting that Dobson and/or ozonesonde data were also less accurate during the first period.

The correlation coefficient is not the appropriate characteristic to describe uncertainties of individual data sources since it also depends on the variability of ozone itself, which is low in the wintertime polar vortex. As total ozone data from several sources are available, information of the instrument uncertainties and ozone variability can be derived by comparing data from individual sources (Grubbs, 1948; Fioletov et al., 2006; Toohey and Strong, 2007; Zhao et al., 2016): A measurement result ($M$) is the sum of the true value ($X$) and an error ($e$). Supposedly, the two instruments measure the same parameter $X$,

but with different errors $e_1$ and $e_2$. If we assume that the measured value and the errors are independent and the errors of different instruments are not correlated, then the results of their measurements ($M_1$ and $M_2$) can be used to estimate the variance of $X$, $e_1$ and $e_2$:

$$\sigma^2(X) = \tfrac{1}{2}\,(\sigma^2(M_1) + \sigma^2(M_2) - \sigma^2(M_1 - M_2)),$$

$$\sigma^2(e_1) = \tfrac{1}{2}\,(\sigma^2(M_1) - \sigma^2(M_2) + \sigma^2(M_1 - M_2)),$$

$$\sigma^2(e_2) = \tfrac{1}{2}\,(\sigma^2(M_2) - \sigma^2(M_1) + \sigma^2(M_1 - M_2)).$$

Equations above can be used to estimate the standard deviation of instrument errors (or instrument uncertainty) and the standard deviation of ozone variability from pairs of coincident Dobson, Brewer, and MERRA-2 data points (ozonesonde data are too sparse). The results for daily and monthly values are shown in Table 1. The values are given for two periods that correspondent to the two MERRA-2 data sources (SBUV and Aura). The 1980-1984 interval of rapid ozone changes was excluded from the

calculations.

       The MERRA-2 uncertainties are lower in 2005-2022 than for the first period suggesting that addition of MLS data improve the reanalysis. The uncertainties in Dobson data appear to be larger than those for Brewer and MERRA-2, but this is because we compared daily averages. Unlike the Brewer and MERRA-2 that provided multiple measurements throughout the day, the Dobsons provided only one value. The uncertainties of Dobson and Brewer daily mean FM values are 6-11 DU or

about 2.5%-4%. The estimated ozone variability is relatively low, about 15 DU for daily averages and 10 DU for monthly values or about 6% and 4%, respectively. Therefore, the wintertime total ozone levels can be established from a relatively limited number of measurements.

       Statistics of mean wintertime total ozone (i.e., one value per year calculated as an average of all wintertime data in that year) are available from Table 2. The values are given in DU for two time intervals discussed above (i.e., 1985–2004 and

2005–2022). Figure 5 (top) shows the time series on wintertime total ozone values for 2005–2022, i.e., for the period of the most accurate ozone values. Individual datasets capture the main features of year-to-year variability on wintertime ozone, although there are some differences likely caused by instrumental issues. The correlation coefficients between Dobson, Brewer, and ozonesonde values and MERRA-2 were in the range from 0.74 to 0.85. The correlation of their average with





MERRA-2 is even higher, 0.9 as instrumental noise and sampling issues are partially canceled out. The standard deviation of the difference between MERRA-2 and Dobson, Brewer, and ozonesonde wintertime values is between 5 DU and 7 DU (2%-3%). It is even lower (3.7 DU) for the difference between MERRA-2 and the average of the Dobson, Brewer, and ozonesonde wintertime values.

As mentioned, the number of Dobson, Brewer, and ozonesonde measurements is very limited in the wintertime. To illustrate the sampling issues, Figure 5 (bottom) shows time series of MERRA-2 data for the same period taken at the time of Dobson, Brewer, and ozonesonde measurements as well as the complete MERRA-2 record. In other words, MERRA-2 data were re-sampled at the time of the actual Dobson, Brewer, and ozonesonde measurements and then compared with a complete MERRA-2 record. Although the Dobson, Brewer, and ozonesonde measurements are available only for 15%–20% of all days

in April-August, MERRA-2 sampled on days and times of their measurements can successfully reproduce mean wintertime values calculated from the continuous MERRA-2 record. The standard deviation between them is only 2.5–3.5 DU (1%–1.5%), while their average of all these sampled data has the standard deviation from the complete MERRA-2 record of only 2.2 DU (~ 0.9%).

## 4 Long-term changes in wintertime total ozone

### 4.1 Time series

There were a total of 258 daily FM measurements by Dobsons nos. 80 and 82 in April-August in 1964-1980. The average of these measurements is $280 \pm 3.2$ DU (2-σ level), that can be used as a benchmark for the pre-1980s ozone and then the deviation from that level can be estimated. Figure 6 (top) shows the deviations from this pre-1980 level for Dobson, Brewer, ozonesonde and MERRA-2 total ozone. Each symbol on the plot represents a five-year average. There is a clear decline from the pre-1980s

level, and the ozone values were about 12% below that level after the mid-1980s. As discussed above, the differences between individual datasets are 2%-3%, so such large deviations can be reliably measured by several independent datasets. Moreover, the MERRA-2 record started in 1980 and the deviation from the 1980 level and the values in 2000 is about 12%. So, a 12% decline can be seen independently from both the Dobson record and MERRA-2 total ozone (the SBUV period).

        The year-to year variability of wintertime polar ozone (Table 2) in recent years is relatively small compared to that

12% decline. The standard deviations of the wintertime values are only 3%-4% for 2005–2022, so a 12% decline corresponds to from 3 to 4 standard deviations.  Moreover, the maximum values over that period from all four data sources were at least 16 DU (or more than 6%) below the benchmark value of 280 DU.

        This observed 12% decline of wintertime polar ozone is much larger than a long-term decline from the pre-1980s levels over southern middle and high latitudes, where the decline is about 5% (e.g., Weber et al., 2022; WMO, 2022). This is

further illustrated in Figure 6 (bottom), which shows wintertime deviations from the pre-1980 level over low and high southern latitudes from two independent data sets of zonal mean values: the WOUDC ground-based data set (Fioletov et al., 2002) and



the merged SBUV data set (Frith et al., 2014). It is not unexpected because the trend estimates for the southern hemisphere for the period from the late 1970s to the late 1990s show an increase in the negative trend magnitude going toward the pole (e.g., Fioletov et al., 2002; Vyushin et al., 2007; Weber et al., 2022). The wintertime long-term decline over the South Pole is probably the largest long-term ozone decline aside from the springtime Antarctic ozone depletion.

5        Figure 7 (the left panel) shows seasonal mean total ozone over the South Pole for four seasons (January-February, April-August, and October-December) from the four data sources. The October–December ozone shows the largest decline, but the year-to-year variability is also the largest due to the variability of the extent and duration of the ozone hole. The interpretation of these data clearly requires additional proxies (e.g., de Laat et al., 2015). January-February data show a smaller decline and smaller variability than the data in October–December. In October-February, SBUV and OMI total ozone are available for MERRA-2 data assimilation and Dobson and Brewer can take the most accurate DS measurements, so the agreement between the different data sets is very good (as discussed in Section 3). Section 3 also demonstrates that the correlation between Dobson and MERRA-2 data is not very high in April–August during the MERRA-2 SBUV period and this can be also seen in Figure 7. The agreement between the four data sets in April–August has become much better after 2005, i.e., when MERRA-2 started to use MLS data.

## 4.2 The EESC fit

The long-term ozone decline is caused by an increase of ozone depleting substances (ODSs) in the stratosphere (WMO, 2018; 2022). The ODSs concentration is often described by equivalent effective stratospheric chlorine (EESC) (Newman et al., 2007) and EESC is used as a proxy of long-term changes in total ozone (e.g., Fioletov and Shepherd, 2005; Stolarski et al., 2006; Wohltmann, 2007; Vyushin et al., 2007; de Laat et al., 2015). Although the results of the EECS-based estimates for the ozone recovery trend assessment should be interpreted with caution (Kuttippurath et al., 2015), we can use them to verify how well the EESC curve describes the observed long-term ozone changes. For EESC we used a version with an age of air 5.5 years that corresponds to the polar stratosphere (Newman et al., 2007). The fitting results of Dobson data by the EESC function for the three seasons are shown in Figure 7 (the right column). Fitting was done separately for each month and then the fitting results were averaged based on Dobson data availability to form the seasonal means. The fitted curve followed April-August Dobson values well and the analysis of the residuals (not shown here) confirmed that.

The main advantage of studying long-term changes in wintertime ozone is that, unlike all other months, the long-term changes in April-August are very uniform. This can be illustrated by the EECS fits. Figure 8 shows the fitting results of Dobson (Figure 8 top) and MERRA-2 (Figure 8 bottom) data where the values for each month were fitted separately. The fitting results for April-August are very similar (and different from the fitting results for all other months). This similarity means that the April-August data can be lumped together for long-term change studies. It should be mentioned that if data for August 21-31 were included into the August record, the fitting curve for August would noticeably deviate from the April-July fitting curves.



As the next step, the average of all data for the period from April 1 to August 20 of each year were calculated as was the wintertime total ozone using the EESC curve. The results are shown in Table 3 in the form of ozone values in different years, estimated from the fit. Dobson and MERRA-2 data show a decline from the pre-1980s level to 2001 (the maximum of the polar EESC curve) of 12% and 11%, respectively, i.e., similar to what was seen on the deviation plot (Figure 6). There is

5 an additional decline of about 2% from 1964 to 1980.

It is interesting to note that the April-August fitting results for Dobson data and MERRA-2 are very similar, although the first 15 years of the Dobson record are not available from MERRA-2. Moreover, the EESC fit is not very different even if we estimate it from the MERRA-2 SBUV period only (i.e., limit the data to 1980–2004). The decline from the pre-1980s level is about 10%. This can be used as an argument that long-term wintertime ozone changes indeed follow the EESC curve. The

10 uncertainties of the EESC fit are given in the bottom line of Table 3 as the ratio between the parameter of the EESC fit to its standard deviation. For the Dobson record, the ratio is 7.5, so the decline from the pre-1980s has the two-sigma uncertainty of 3.3%, so the difference in estimated declines for 1964–2022 from Dobson data and for 1980–2004 from MERRA-2 data are within the uncertainty. The uncertainty of MERRA-2 – based estimates are larger (the ratios are smaller) due to a shorter time interval.

**4.3 On ozone recovery in the wintertime**

Detection of ozone recovery is an important current research topic (e.g., Steinbrecht et al., 2018, Weber et al., 2022). From the EESC changes, it is expected that the magnitude of positive recovery trends are only one-third of the magnitude of the magnitude of negative ozone decline trends. A large natural ozone variability makes the detection of a such small rate of recovery complicated. In addition, the total ozone recovery may be also inhibited by ozone decline in the lower stratosphere

(Ball et al., 2018). The South Pole wintertime total ozone record was examined to detect the recovery in two ways: from the EESC fit and as a linear trend in the wintertime values. The results are summarized in Table 4 for two time intervals. The 2001-2022 interval corresponds to the declining part of the EESC curve, while the 2005-2022 interval corresponds to the Aura part of MERRA-2. From the EESC fit, the increase is 1.3%-1.6% per decade. Dobson and ozonesonde data for 2001-2022 show similar values of the linear trend, although the trend 1-σ uncertainties are as large as the trend itself. However, the

MERRA-2 trend is nearly half that of the Dobsons and ozonesondes, probably due to a lower quality of MERRA-2 data from the SBUV period. For the 2005-2022 period, data from Dobsons, ozonesondes and MERRA-2 show a similar decline, however, the magnitude of that trend is smaller than that for 2001-2022. The trend from Brewer data is almost zero, but it cannot be compared with the mentioned Dobson and ozonesonde trends since the Brewer record started only in 2008.

These trend uncertainty estimates reflect large ozone fluctuations shown in Figure 5. They are likely caused by

30 dynamical factors related to the formation and strength of the polar vortex. Therefore, the uncertainties can be reduced by adding proxies that are related to these factors. However, such analysis is outside the scope of this study.



## 5 Discussion and conclusion

The Antarctic polar vortex creates unique chemical and dynamic conditions when the stratospheric air over Antarctica is isolated from the rest of the stratosphere. The vortex is formed in late autumn, and it breaks up, typically, in October–December. The sunrise and extremely cold temperatures create favorable conditions for rapid ozone loss after sunrise, however, for a five-month period from April until late August, stratospheric ozone within the vortex remains largely unchanged. Such prolonged stable conditions within the vortex make it possible to estimate the total ozone levels there from sparse wintertime ozone observations at the South Pole.

The available records of focused Moon (FM) observations by Dobson and Brewer spectrophotometers at the South Pole Station (for the periods 1964–2022 and 2008–2022, respectively) as well as integrated ozonesonde profiles (1986–2022) and MERRA-2 reanalysis data (1980-2022) were used to estimate the total ozone variability and long-term changes. Some adjustments were applied to the original data to make the data records consistent. No adjustment was applied to the Dobson record, only three unrealistic values were removed in 2013. Ozonesonde data were decreased by 2% to match the Dobson values. MERRA-2 data have about 10% systematic bias between the wintertime values during the SBUV period (1980–2004) and OMI/MLS period (2005–2022). To remove this bias and make the data consistent with the Dobson record, wintertime MERRA-2 data for 1980–2004 were increased by 8.5% and data for 2005–2022 were decreased by 1.7%.

While Dobsons typically report only one measurement per day, Brewers provide a nearly continuous record of total ozone measurements when the Moon is full and lunar zenith angles are small enough. In general, Brewers report an average wintertime total ozone level that is similar to that from the Dobson. There is, however, a major issue with Brewer FM data: they are overestimating ozone by 10%-15% when the slant path is high (greater than 1000 DU). This is likely related to the instrument's performance when the lunar radiation is low.

Although Dobson, Brewer and ozonesonde measurements are sparse, they can be used to accurately estimate wintertime ozone. The sampling effect was estimated by comparing wintertime ozone values calculated from continuous MERRA-2 data with MERRA-2 data sampled only at the time of Dobson, Brewer and ozonesonde measurements. This comparison demonstrated that the wintertime values can be estimated from such MERRA-2 re-sampled data with a standard deviation of 2.5–3.5 DU (or 1%–1.5%), while the average of all these re-sampled data has the standard deviation from the complete MERRA-2 record of only 2.2 DU (~ 0.9%).

The wintertime ozone variability over the South Pole is low. The estimated standard deviation of the ozone variability is 13–15 DU for daily values and 7–10 DU for monthly means. The standard deviation of annual mean wintertime ozone is about 10 DU, although this number is inflated because it includes some instrumental errors. This number is small compared to the 30 DU (11%–12%) ozone decline from the average of about 280 DU prior to 1980 and the present level of about 250 DU.

The wintertime total ozone variations over the South Pole support the statement that the changes that are expected agree with the shape of the EESC curve. From the EESC fit, the decline from the pre-1980s level to 2001 (the maximum of the polar EESC curve) is about 12%. There is an additional decline of about 2% from 1964 to 1980. From the EESC fit, the

expected ozone increase rate after 2001 is 1.3%–1.6% per decade. Although the variability in wintertime ozone is not very high, it is still difficult to find a statistically significant positive ozone recovery trend. Dobson and ozonesonde data demonstrate 1.3%–1.6% positive trend for the period of the EESC curve maximum (2001-2022), but the 1-σ trend uncertainties are as large as the trend itself. MERRA-2 data for the same period show only half of the observed trend, perhaps because of large
uncertainties during the MERRA-2 SBUV period.

Wintertime polar ozone is affected by all the factors contributing to the changes in the ozone layer, probably, to the largest extent. Rapid ozone destruction on polar stratospheric clouds in the springtime Antarctic vortex affects ozone levels in subsequent months everywhere in the southern hemisphere, but its impact on the polar ozone should be at least as strong as anywhere else. A decline in ozone due to gas-phase ozone destruction from ODSs is probably the largest, since the time for an
air parcel to travel from the tropics to high latitudes is the longest. As a result, the decline in wintertime polar ozone is probably the largest long-term ozone decrease aside from the springtime Antarctic ozone depletion. Possible changes in the Brewer-Dobson circulation in the southern hemisphere would also likely have a larger impact over the South Pole than over the lower latitudes. The wintertime ozone values over the South Pole during the last 20 years were about 12% below the pre-1980s level, i.e., the decline there was nearly twice that of over southern midlatitudes. Thus, wintertime ozone values in the polar vortex
can be used as an indicator to diagnose the state of the ozone layer.  It is also important to stress that such diagnostics requires data only from one station.

*Data availability.* Brewer data at the South Pole Station are available from the World Ozone and UV Data Centre ([https://woudc.org/](https://woudc.org/) ). Dobson and ozonesonde data at the South Pole Station are available from NOAA's Global Monitoring
Laboratory ([https://gml.noaa.gov/ozwv/](https://gml.noaa.gov/ozwv/)).  MERRA-2 data are available from NASA's Global Modelling and Assimilation Office ([https://gmao.gsfc.nasa.gov/reanalysis/MERRA-2/](https://gmao.gsfc.nasa.gov/reanalysis/MERRA-2/) ).

*Author contributions.* VF analyzed the data with help from XZ and IA and prepared the manuscript, with critical feedback from all co-authors. CTM installed the Brewer instrument at the South Pole Station. MB, RS, AO, VF, TM and SCL operated
and managed the South Pole Brewers. IP, BJJ, PC, JB, KM, and GM operated the South Pole Dobson and performed the ozonesonde measurements.

*Acknowledgements.* We also would like to acknowledge Christopher Osburn, Lunar Outreach Services, for developing a free software package for the Moon phase calculations used in this study. We also acknowledge the logistics support in Antarctica
provided by the National Science Foundation, Office of Polar Programs and Amy Cox for her support in setting up the Brewer instrument



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





**Table 1.** Standard deviation for monthly and daily values, estimated uncertainties and total ozone variability (DU).

|  | Daily values | | Monthly values | |
|---|---|---|---|---|
|  | 1985–2004 | 2005–2022 | 1985–2004 | 2005–2022 |
| Dobson FM standard deviation | 22 | 19 | 19 | 15 |
| Brewer FM standard deviation | - | 15 | | 12 |
| MERRA-2 standard deviation | 21 | 15 | 17 | 10 |
| Dobson FM uncertainty | 18 | 11 | 17 | 10 |
| Brewer FM uncertainty | - | 6 | | 5 |
| MERRA-2 uncertainty | 17 | 5 | 16 | 5 |
| Ozone variability | 13 | 15 | 7 | 10 |

The uncertainties are estimated from two pairs of data sources, e.g., the Dobson uncertainty can be estimated from the Dobson & Brewer
5 and Dobson & MERRA-2 pairs and their average is shown in the table. The ozone variability can be estimated from three pairs and their
average given in the table.

**Table 2.** Statistics of the mean wintertime ozone for 1985-2004 and 2005-2022 in DU.

|  | Mean | Standard deviation | Minimum | Maximum |
|---|---|---|---|---|
| | | 1985–2004 | | |
| Dobson | 250 | 13 | 229 | 269 |
| Ozonesonde | 249 | 12 | 230 | 271 |
| MERRA-2 | 251 | 13 | 226 | 282 |
| | | 2005–2022 | | |
| Dobson | 247 | 10 | 229 | 263 |
| Ozonesonde | 248 | 10 | 224 | 264 |
| MERRA-2 | 248 | 7.6 | 229 | 260 |
| Brewer* | 244 | 7.6 | 230 | 256 |

15  *Brewer data are available only for the period 2008-2022





**Table 3.** Total ozone values (in DU) in 1964, 1980, 2001, and 2022 estimated from the EESC fit for Dobson and MERRA-2 data. The overall decline in 2022 from the 1964 level and the pre-1980s values and the ratio of the EESC-related fitting parameter to its standard deviation.

| Year | Dobson (1964–2022) | MERRA-2 (1980–2022) | MERRA-2 SBUV (1980–2004) |
|---|---|---|---|
| 1964 | 286 | 281 | 280 |
| 1980 | 271 | 268 | 268 |
| 2001 | 244 | 245 | 247 |
| 2022 | 252 | 252 | 253 |
| Decline from 1964 (%) | 14.7 | 12.8 | 11.8 |
| Decline from pre-1980s (%) | 12.4 | 10.7 | 9.9 |
| Fit parameter to its uncertainty ratio | 7.5 | 3.5 | 2.5 |

**Table 4.** The ozone recovery trends and their standard deviations (in brackets) for 2001-2022 and 2005-2022 from different data sources. The values are given in % per decade and as a total change in %.

| | Dobson | MERRA-2 | Brewer* | Ozonesonde |
|---|---|---|---|---|
| | | 2001–2022 | | |
| Trend (% per decade) | 1.6 (1.5) | 0.7 (1.1) | | 1.3 (1.3) |
| Total increase (%) | 3.4 (3.3) | 1.4 (2.4) | | 2.4 (2.4) |
| EECS trend (% per decade) | 1.6 (0.2) | 1.3 (0.4) | | |
| EESC total increase (%) | 3.6 (0.5) | 3.0 (0.9) | | |
| | | 2005–2020 | | |
| Trend (% per decade) | 1.1 (1.9) | 0.8 (1.4) | -0.1 (1.9) | 0.9 (1.9) |
| Total increase (%) | 2.0 (3.4) | 1.4 (2.5) | -0.1 (3.4) | 1.6 (3.4) |
| EESC trend (% per decade) | 1.6 (0.2) | 1.3 (0.4) | | |
| EESC total increase (%) | 2.9 (0.4) | 2.4 (0.7) | | |

15   *Brewer data are available only for the period 2008–2022







**Figure 1.** (top 4 panels) Total ozone time series from Dobson, Brewer, MERRA-2, and ozonesonde data for 1963–2022 as indicated on the plot. Each dot represents a daily mean value. The vertical lines correspond to Dobson and Brewer instrument changes. The instrument serial numbers are also shown. (bottom panel) The solar (red) and lunar (blue) zenith angles as a function of time. Note long-term variations of maximum lunar zenith angles between 63° and 73°.



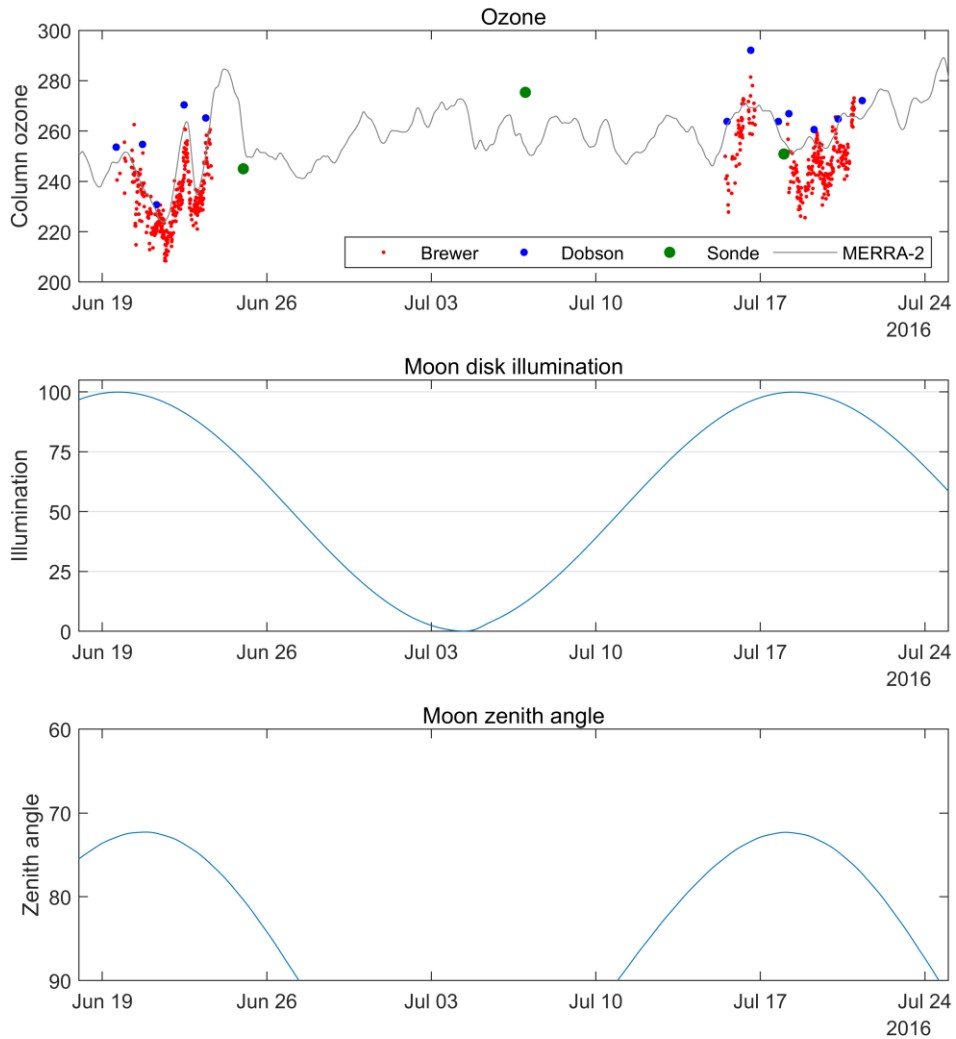

**Figure 2**. (top) An example of wintertime ozone time series: Total ozone from Dobson (the blue dots), Brewer (the red dots), ozonesonde (the large green dots), and MERRA-2 (the gray line) data for June 18 – July 24, 2016. (middle) The lunar disc illumination and (bottom) zenith angle for the same period.



**Figure 3.** (left) Differences between MERRA-2 data and Dobson (blue), Brewer (red), and ozonesonde (green) total ozone in percent as a function of the year (upper panels) and month of year (bottom panels) for original data. Comparison was done for two seasons: April-August and October-February and for two intervals: 1980–2004 and 2005–2022. (right) The same, but for the data after adjustments. The vertical lines indicate the period from April to August. The error bars correspond to two standard errors of the mean.





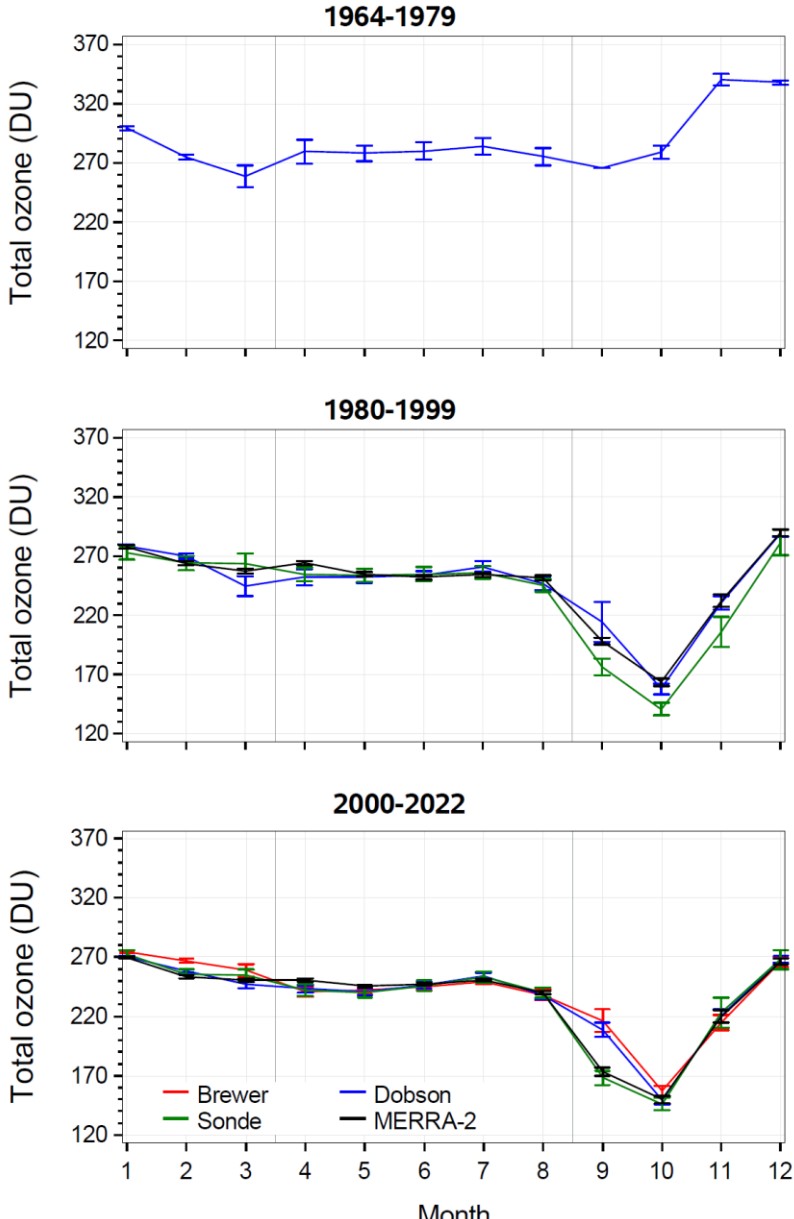

**Figure 4.** Total ozone annual cycle from Dobson (blue), Brewer (red), ozonesonde (green), and MERRA-2 reanalysis (black) data for three intervals as indicated on the plot. The vertical lines indicate the period of the stable ozone in the vortex from April to August. The error bars correspond to two standard errors of the mean. The differences in September-December are caused by the sampling bias: The number of Dobson and Brewer measurements in March and September is very limited they are missing in the second half of September. Ozonesonde flights were more frequent during when the ozone hole was over SPO. Note that there is no sampling bias in April-August.





**Figure 5.** (top) Mean wintertime ozone for 2005-2022 from Dobson (blue) and Brewer (red) daily values, ozonesonde (green) total ozone and MERRA-2 reanalysis (black). The average of Dobson, Brewer, and ozonesonde data is shown by the orange line. (bottom) The same as above, but with MERRA-2 data coincident with Brewer and Dobson observations and ozonesonde flights used instead of the actual measurements. Note that the pre-1980s level is 280 DU.





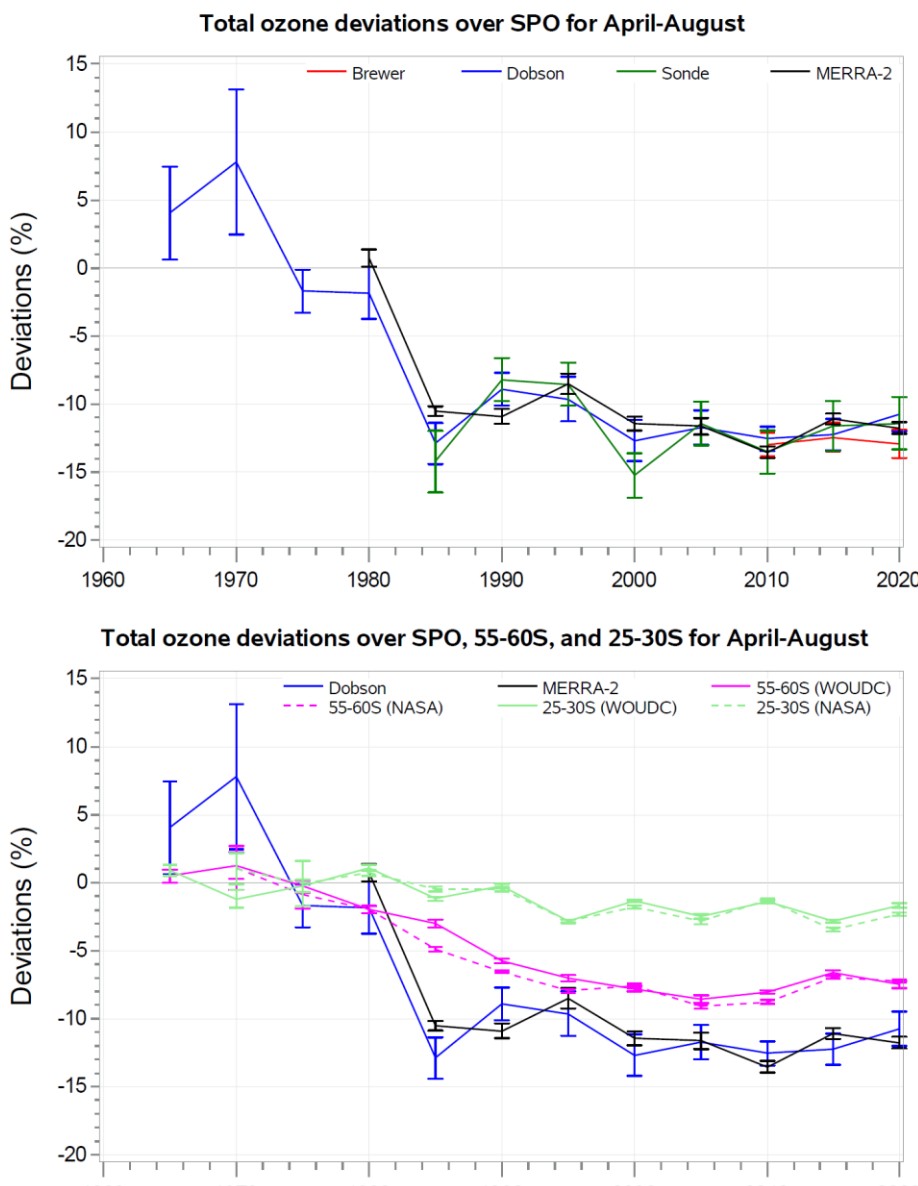

**Figure 6**. Mean total ozone deviations from the pre-1980 level for April-August in percent. (top) Deviations over the South Pole base on Dobson (blue) and Brewer (red) FM measurements, integrated ozonesonde profiles (green), and MERRA-2 reanalysis (black) data. Data were adjusted as discussed in the text. (bottom) Deviations over the pole from Dobson and MERRA-2 data (the same as on the top panel) and ozone deviation from the pre-1980 level over 25°S-30°S (magenta) and 50°S-60°S (green) estimated from NASA merged satellite data set (dashed lines) and WOUDC ground-based data set (solid lines). Each symbol represents a 5-year average, the error bars correspond to two standard errors of the mean.



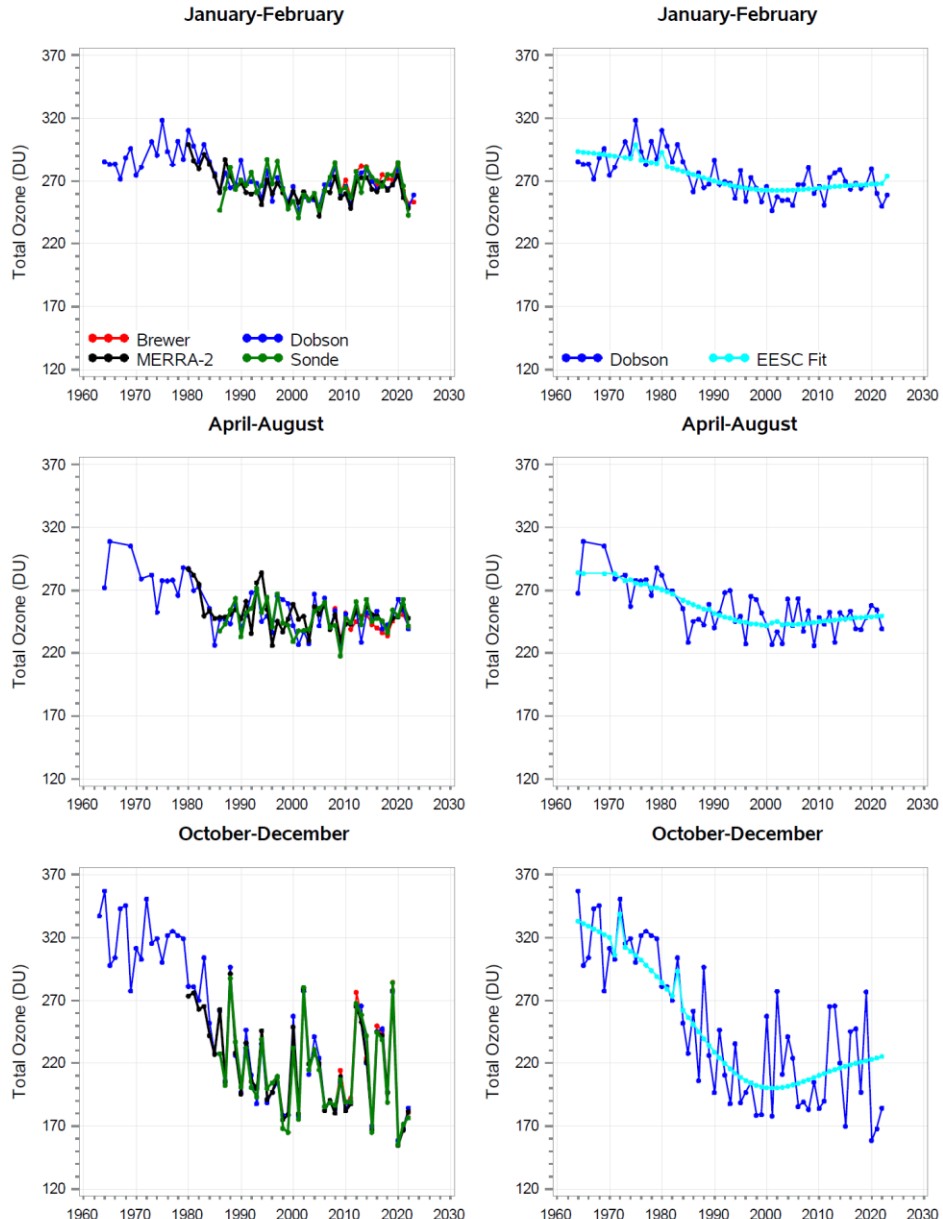

**Figure 7.** (left) Seasonal mean total ozone over the South Pole from Dobson (blue) and Brewer (red) FM measurements, integrated ozonesonde profiles (green), and MERRA-2 reanalysis (black) data for three seasons as indicated on the plot. (right) Dobson seasonal mean ozone (blue) and the fit (cyan) of Dobson data by the equivalent effective stratospheric chlorine (EESC) curve. Fitting was done separately for each month and then the fitting results were averaged based on Dobson data availability to form the seasonal means.



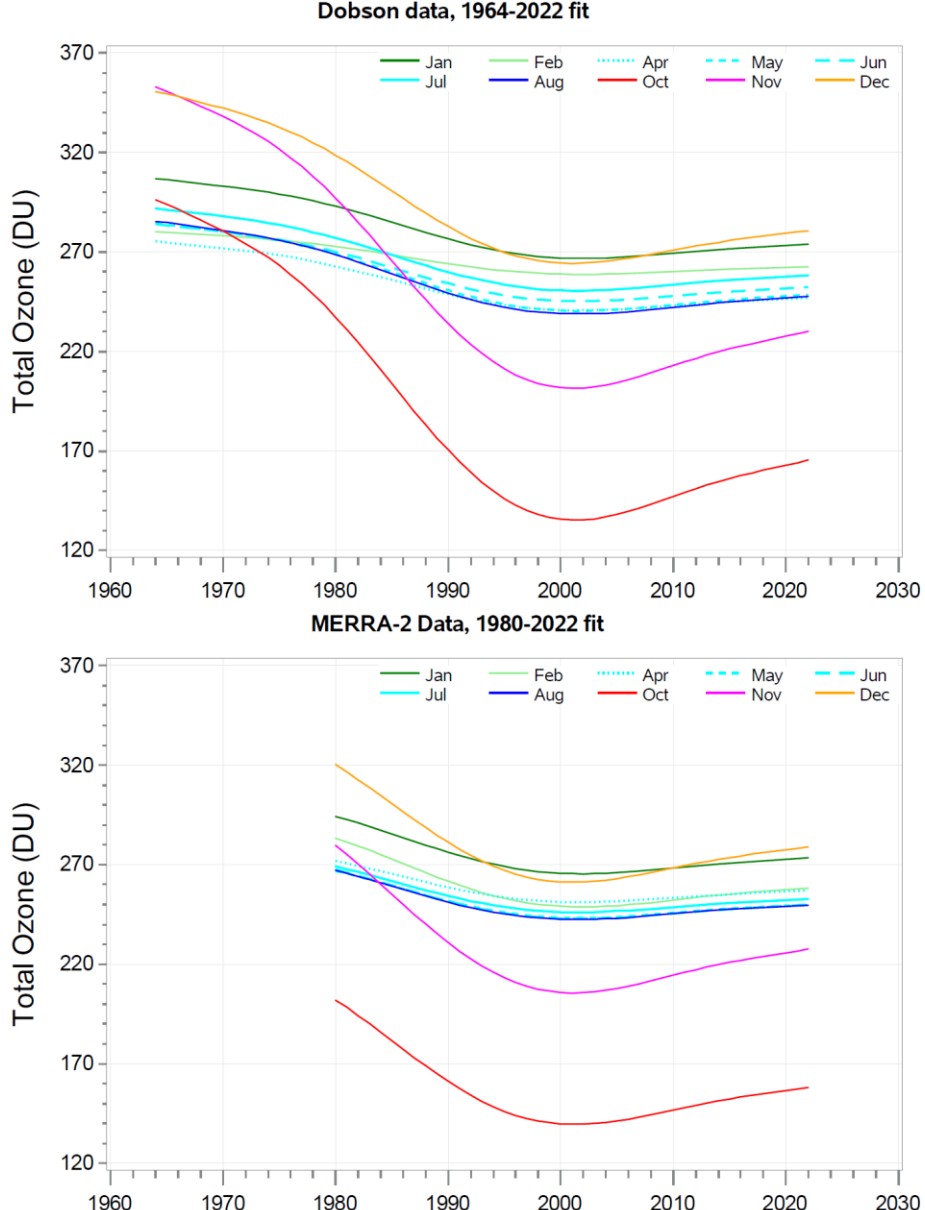

**Figure 8.** (top) Fits of Dobson data in different months (as indicated on the plot) by the EESC curve using 1964-2022 data. (bottom) the same, but for MERRA-2 data for 1980-2022. Although fitting was done individually for each month, the difference between April and August EESC curves (cyan and blue colors) are within 10–15 DU, suggesting that long-term changes in the polar vortex total ozone are uniform for all wintertime months. Note that in August, only the first 20 days were used for the fit.





## Appendix A: The lunar disk illumination and lunar zenith angles

The combination of the Earth rotation and the Moon rotation around the Earth create a peculiar pattern of the lunar disc illumination and lunar zenith angle distribution as shown in Figure A1 for two years. Figure A2 shows the solar and lunar zenith angles for the periods of high and low moon elevation above the horizon. Since only conditions with the moon disk illumination greater than 50% and zenith angles less than 76° for Brewer and ~80° for Dobson are suitable for measurements, there are only 5-6 short periods per winter, when such measurements can be performed.

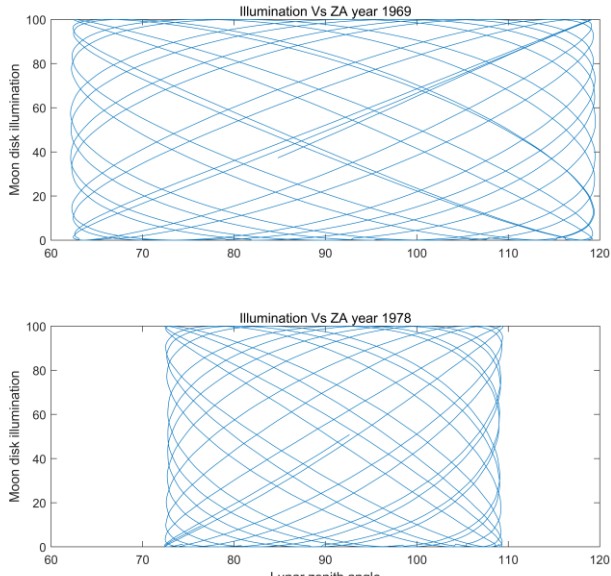

**Figure A1.** The distribution of the lunar disk illumination and lunar zenith angle values in (top) 1969 and (bottom) 1978.



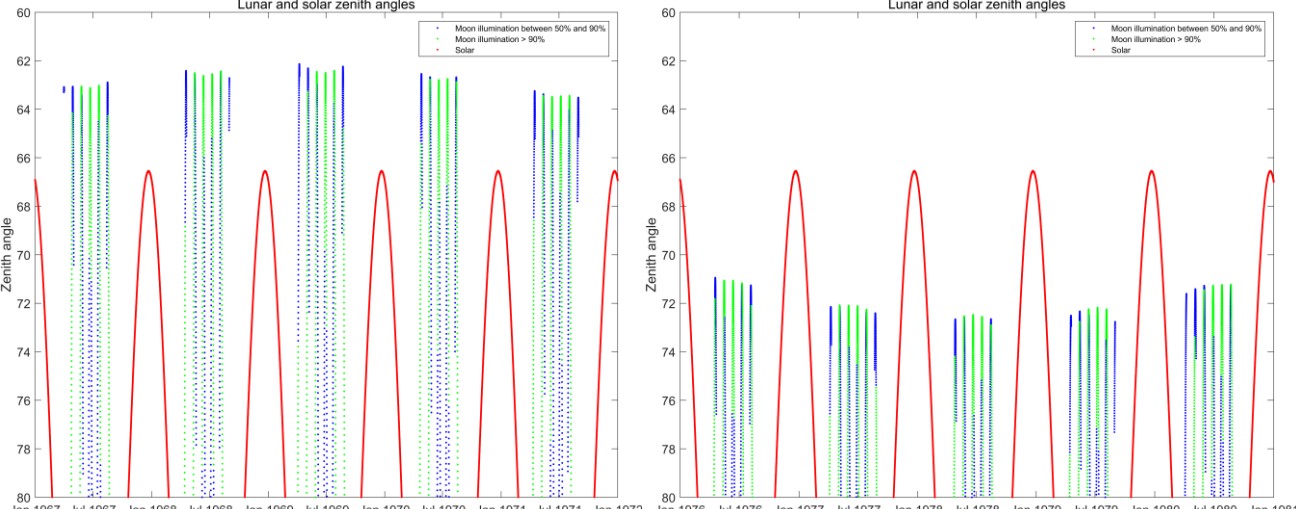

**Figure A2.** Plot of the solar (red) and lunar zenith angle (green and blue) as a function of time for two time intervals (and 1967-1971 and 1976-1980) that correspond to periods of high and low moon elevation above the horizon. Each dot corresponds to one hour. Blue dots correspond to the lunar disk illuminations between 50% and 90%, green dots correspond to the lunar disk illumination above 90%.



## Appendix B: Brewer data corrections

We found that Brewer measurements overestimate ozone when the lunar radiation intensity is low. "Signal 320", i.e., the natural logarithm of the number of photon counts per second at Brewer slit 5 (approximately 320 nm), adjusted for the dead time, the number of "dark" counts and instrument temperature response (Kerr, 2010) was used here to assess the lunar radiation
5   intensity. Figure B1 shows the difference between Brewer and MERRA-2 ozone data as a function of Signal 320. For low lunar radiation intensity, Brewer ozone values are higher than MERRA-2 by 10-15% for Brewer no. 085 and by 5-10% for Brewer no. 21, although both instruments show near-zero differences for Signal 320 equal to 11. Such dependence of the difference on the moon radiation intensity is probably related to nonlinearity of the Brewer photomultiplier sensitivity at low signals. The 320 nm is the longest wavelength used in the Brewer FM and DS ozone retrieval algorithms (besides the other
10  three shorter wavelengths) and the ozone absorption is low at that wavelength. Therefore Signal 320 it is practically not affected by the ozone slant column and all ozone measurements can be grouped by Signal 320.

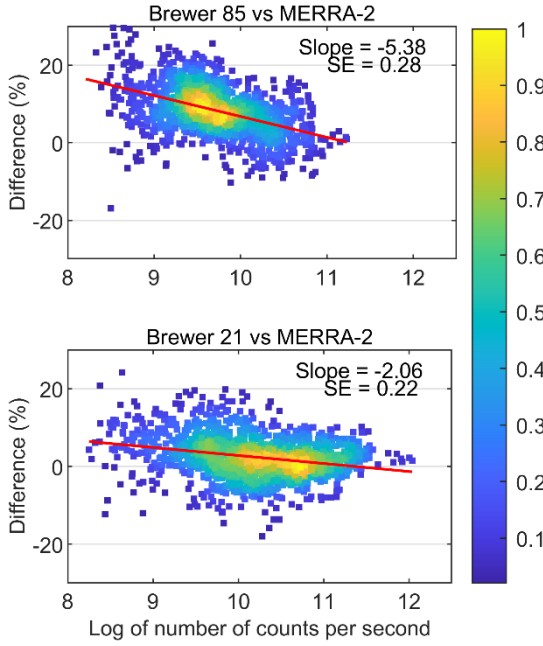

**Figure B1.** Scatter plots of the difference between Brewer FM total ozone and MERRA-2 reanalysis as a function of is the natural logarithm of the adjusted number of photon counts per second at Brewer slit 5 (approximately 320 nm). The number of counts was adjusted for the dead time, the number of "dark" counts and instrument temperature response. The slope of the linear fit and the standard error (SE). of the slope are also shown. The color scale shows the normalized density of the points.

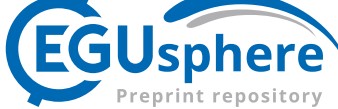



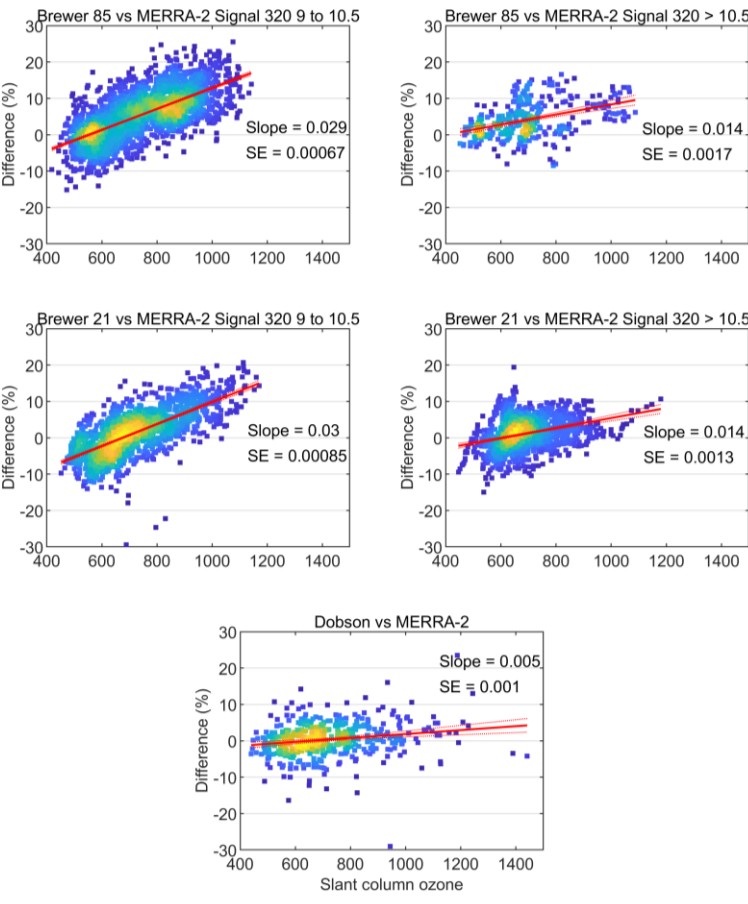

**Figure B2.** Scatter plots of the difference between Brewer and MERRA-2 total ozone as a function of slant column in DU. The best fit linear regression line, its slope value, and the standard error of the slope (SE) are also shown. The difference is plotted for Brewers 21 and 85 as indicated and for two ranges on Signal 320 values: from 9 to 10.5 and greater that 10.5. The color scale shows the normalized density of the points. A similar plot for all Dobson data is show for comparison.

Comparison with MERRA-2 was used to evaluate possible biases in Brewer data since MERRA-2 does not depend on lunar radiation. Figure B2 shows the difference between Brewer and MERRA-2 total ozone as a function of the slant column for the values of Signal 320 between 9 and 10.5 (the left column) and above 10.5 (the right column). A similar plot for the Dobson measurements is also shown. The slope in increasing with decline of the Signal 320. We removed all data for the Signal 320 less than 9 because the faction of such measurements is small and the bias in ozone values is large. The data with Signal 320 greater than 10.5 show some dependence of the difference on the slant column, but most of such data correspond to slant columns under 800 where the difference is small. However, if we just discard all data with Signal 320 less than 10.5, the number of days with FM measurements is reduced by 50% for Brewer 21 and by 80% for Brewer 85. For this reason, we



applied an empirical correction ($\Delta O_3 = (SlantColumn - 750\,\text{DU}) \times 0.062$) to remove a linear trend in the difference with MERRA-2 as a function of the slant column if the Signal 320 is between 9 and 10.5. This correction has completely removed the dependence of the difference on Signal 320 for Brewer no. 021, but for Brewer no. 085. As Figure B1 shows, the lunar radiation dependence effect was larger for Brewer no. 85 than for Brewer no. 21, while the suggested correction was the same for both

5 Brewers. We applied another correction for Brewer no. 085 that was 0 for the Signal 320 greater than 10.5 and linearly decreased from 0 to -4% for Signal 320 declining from 10.5 to 9.

It is important to note that the applied empirical correction did not change the wintertime mean total ozone values for the two Brewer instruments compared to the scenario where all data with Signal 320 less than 10.5 were discarded. For the latter scenario, the mean wintertime ozone values measured by Brewers 85 (in 2008-2014) and 21 (in 2015-2022) were 244

10 DU and 241 DU respectively, while for the corrected data, they were 244 DU and 242 DU, respectively. Thus, the correction did not change the average ozone level established by the most reliable Brewer FM measurements. The correction also has improved the correlation coefficients between Brewer data and the other data sets. The correlation coefficients of Brewer daily values with Dobson, MERRA-2, and ozonesonde were 0.59, 0.71, and 0.65, respectively, for the original data and 0.73, 0.8, 0.74 for the adjusted data.

