# Peer review of "Total ozone variability and trends over the South Pole during the wintertime"

_EGUsphere, 2023_

## Author Comment (AC1)

Review of the manuscript with the title "Total ozone variability and trends over the South Pole during the wintertime" by Fioletov et al.

The manuscript studies total ozone measurements at the South Pole during winter time. Dobson, Brewer, ozone soundings and MERRA-2 data are used. The study includes quality assurance of the data as well as uncertainty estimates. This is the first time the analysis is done for such a long period in the winter time, as previous studies are more focused on springtime Antarctic ozone loss. The study brings new information on total ozone trends in the Southern polar area and adds valuable information to global ozone trends at those latitudes. The manuscript is well-written and includes appropriate references. The authors should address the general comments and the more specific comments before publication of the manuscript.

We would like to thank the reviewer for his/her favorable comment.

General Comments:

My main concern is related to the correction of MERRA-2 data which is based on only one correction factor over the whole winter when analysing the MERRA-2 SBUV period. Why not use a correction factor per month, as suggested by the Figure 3? Please see also the specific comments. The second concern is the use of the EESC fit. For a reader not familiar with the methodology the text is rather confusing. This needs to be clarified.

There are two reasons why a single correction was applied instead of corrections for individual months. First, we preferred to change the data as little as possible to keep the data sources independent. Second, most of our results (e.g., Figure 6) are related to wintertime averages. Since MERRA-2 data do not have any gaps, corrections for individual month would have the same impact on these averages as a single correction. In addition, the statistical uncertainty of a single correction factor is less that uncertainties of monthly correction factors. We added this explanation to the text.

Indeed, monthly corrections would improve the agreement between MERRA-2 and other data sets in the SBUV period and would impact some trend results (Figure 8). However, we have other estimates that are based on SBUV and AURA periods separately.

We also added some information about EESC and the data analysis methodology (see response to Specific comments).

Specific comments:

Page 2, Line 5: Polar vortex breakdown time: November-December?

Correct. We changed the sentence and added a reference.

Page 2, Line 22: total ozone declining trend: for which time period? Yearly mean?

We added a sentence about the time period and that the trend here is annual.

Page 3, Line 28: Please specify how the temperature and ozone profiles are used in the retrieval algorithm.

We added a few words for clarification.

Page 4, Line 8 and page 5, line 4: Until which period you can still measure good DS measurements? In the winter, with no Sun, I suppose it is impossible.

DS measurements are not possible between March 22 and September 22 since the Sun is below the horizon. In practice, however, this interval is even larger because DS measurements are not reliable if the Sun is very low. Data are available from October to first days of March. We added this information.

Page 4, Line 9: How do you make the choice of the representative daily value?

We added the following text: "Dobson data processing system selects one of daily observations as representative based on the type of the observation (direct sun or direct moon over the zenith sky), wavelength pair (i.e. AD over CD), height of the sun/moon (i.e. the observation with the smallest zenith angle is preferred) and interference of clouds (clear sky over cloudy condition)."

Page 4, line 30. I don't think Kerr et al. 2010 is a "recent" paper ....13 years old. Maybe better to delete "recent". Please add some sentences on the calibration and harmonization of the Brewer including used corrections to the data, like the use of sl-lamp measurements.

Yes, it is not recent anymore. Corrected. We also added sentences about Brewer calibrations and standard corrections.

Page 6, line 4: For MERRA-2 during the polar night: is the total ozone for the South Pole actually the total ozone at latitude 82 degrees South? Or is it extrapolated somehow from that latitude to the South Pole? Please specify.

MERRA-2 is based on data assimilation. So, ozone measurements everywhere contribute to the MERRA-2 ozone value over the South Pole. The point is that during the SBUV period, there were no ozone measurements for assimilation between 90S and 60S in some months, while for the Aura period this "blank" area was only between 90S and 82S. We added some text to make this statement clear.

Page 6, line 14: "...existence of some systematic seasonal biases..." Does this include any information about the winter (the season you studied)? If yes, please add it to the paragraph.

Wargan et al., (2017) just stated that "While small systematic season-dependent biases exist, the annual cycle, latitudinal structure, and longer-term variability are realistic, and the agreement with the independent data is well within the assumed observation errors." The statement is based on their Figure 4. No additional information was provided.

Page 7, line 1: "This could be related to some horizontal inhomogeneity of the ozone distribution over the pole that led to variations in measured ozone due to changing lunar azimuth angle." I can't follow the sentence. Please clarify. Are the changes real or an artifact caused by changing lunar azimuth angle?

Brewer and Dobson do not measure total ozone exactly over the instrument as their measurements are based on the light absorption by ozone on the line between the instrument and the Sun/Moon. For example, if ozone maximum is at about 20 km and the zenith angle is about 70, then the maximum ozone absorption occurs 60 km from the instrument in the direction to the Sun/Moon. Thus, the Dobson/Brewer measured ozone depends on ozone layer characteristics 60 km or so from the instrument location. As the instrument measures ozone at various azimuth angles (i.e. directions to the Sun/Moon), it, in fact, measures ozone over different areas around the instrument. If the ozone distribution is not horizontally homogeneous, this would yield variations in the measured by Dobson/Brewer total ozone. We added that the measurements in that sentence are related to Brewer.

Page 7, line 31: "There was also a 3% difference..."  Was it the mean difference? Please specify.

Page 7 + Figure 3: Correction of MERRA-2 data for 1980-2004. Why didn't you use monthly mean differences for the correction in April-August? There seems to be a clear month-dependent difference. You can still see a month-dependent difference in the corrected data ranging from -5% to 5%.

Please see the response to General Comments.

Page 9, line 25: "The estimated ozone variability is relatively low, about 15 DU for daily averages and 10 DU for monthly values or about 6% and 4%, respectively." -> Do you mean over the winter period April-August? Please specify.

Yes, everything is related to wintertime ozone. We clarified this in the text.

Page 11, Section 4.2. For a reader not familiar with EESC fit it is very difficult to understand what is fitted to what. You should describe the methodology in more detail, and how did you end up with the fit in the right columns of Fig. 7.

We added additional information and formulas.

page 14, line 10: "A decline in ozone due to gas-phase ozone destruction from ODSs is probably the largest since the time for an air parcel to travel from the tropics to high latitudes is the longest." -> Please open this statement. Is this related to the Brewer-Dobson circulation transporting ozone from the tropics to the pole? Does ODS deplete it the whole way? During which months?:

We clarified that it is due to the Brewer-Dobson circulation in austral spring-summer.

---

## Author Comment (AC2)

This paper reports on ozone changes observed since the 1960s at the Amundsen-Scott South Pole Station (SPO). A particular focus lies in the winter months just before the period of rapid ozone depletion in spring ("ozone hole period"). This time of the year is covered by full moon (FM) measurements by Brewer and Dobson spectrophotometers complemented by regular ozone sondes and MERRA2 reanalysis data. Regular direct sun (DS) measurements by Brewer and Dobsons are also used to contrast the ozone changes in the winter months to other seasons and latitudes. Wintertime ozone declined by about 12% from the pre-1980s until the late 1990s, a larger change than observed at lower latitudes and in other sesaons, except for the ozone hole period. The paper is well written and results are well presented. After adressing some rather minor issues as outlined below, the paper will be well suited for publication in ACP.

First of all, we would like to thank Mark Weber for his favorable comments.

p. 6, l. 15ff: A great part of the paper deals with the adjustments of the various datasets. As a reference for adjusting all data types the long-term Dobson data is used (p. 7, l. 14). For justification only the study by Bernhard et al. (2005) is cited. I think a few more sentences are here needed to explain why the Dobson record is most suitable as a refence dataset here.

The Dobson record was thoroughly reanalyzed by Evans et al., 2017. It was mentioned before in Section 2.1, and we added that reference again here. Also, we reminded that the used Dobson data included the correction for the temperature dependence.

p. 7, l. 32: "... remove that bias for some of the plots". So the corrections are only applied in the plots but not for the data. I find this a bit awkward, why not say simply the data has been corrected, which would be important if the adjusted data im made publicly available (see my later comments on Data Availability)

Sorry for the confusion. Indeed, the data has been corrected. The text is changed to reflect that. Anyways, October-February data are not the main focus of this study.

p. 8, l. 5: I think that the bottom panel is not showing what is described in the main text and figure caption. MERRA2 data are the same as in the top panel, but all other data have been adjusted (not the other way around).

It is probably related to Figure 3 (p.8, l.15). The text is correct. The data (MERRA-2, Brewer, and ozonesondes) without any adjustments were used in four panels on the left side. Plots on the right side show the same analysis, but adjustments applied to MERRA-2, ozonesonde, and Brewer. Major adjustments (~8%) were applied to MERRA-2 data for April-August in 1980-2004 (SBUV period). All other adjustments are relatively minor. We added a few words to the Figure 3 caption to make this clearer.

p. 11, l. 5: "four seasons" --> "three seasons"

Corrected

p. 11., l. 25: not clear what is meant with "analysis of the residuals". Please specify. A fit of only the EESC curve to the data will result in large residuals as the short-term variability is not fitted. Maybe it would be good to show some plots of residuals to make the point here (could be put in the appendix).

We agree. The statement about the residuals is not needed here. The main message from the EESC fit is discussed in the next paragraph: The fitting results for April-August are very similar. We simply removed the statement about the residuals.

Fig. 8 (and other plots): Light blue color lines are dificult to distinguish, in particular with different line styles. The light blue color is not a particular good color for color blind people. I strongly suggest to use a different color. This applies also for the other plots using the same color.

Corrected. The cyan lines were replaced with black lines in Figure 8. We also replaced the cyan line in Figure 7 with a thick gray line.

p. 14, l. 7: "Rapid ozone destruction on polar stratospheric clouds in the springtime Antarctic vortex affects ozone levels in subsequent  months everywhere in the southern hemisphere, but its impact on the  polar ozone should be at least as strong as anywhere else." I find  this sentence a bit awkward. I think what was meant to be said is that  the polar ozone loss (ozone hole) is not affecting the wintertime ozone, so that gas-phase chemistry is only relevant in winter. Apart from ODS gas-phase chmistry some dynamic contributions can not be ruled out as suggested in the main text. I think this should be mentiond here as well.

The polar ozone loss has some impact on SH ozone in summer and fall and may even impact ozone in wintertime. But perhaps the statement is too strong and the point about some dynamic contribution is also correct. We reworded that statement:

*Wintertime polar ozone is affected by all the factors contributing to the changes in the ozone layer, probably, to the largest extent. The contribution from dynamic factors to ozone variations in the polar region is probably similar to that anywhere else in the southern middle and high latitudes.*

p. 14, l. 17: The data at SPO, in particular the adjusted data  should be made available publicly for tracability.

The temperature-corrected Dobson data  are available from https://gml.noaa.gov/aftp/data/ozwv/Dobson/Publications/ .Corrected Brewer FM data used in this study is available from the Supplement to this paper. It is now stated in the Data availability section.

p. 34, l. 4: "in increasing" --> "is increasing"

Corrected